# Foiling Explanations in Deep Neural Networks

**Snir Vitrack Tamam**\*                                                        *snirvt@gmail.com*
*Department of Computer Science*
*Ben-Gurion University, Israel*

**Raz Lapid**\*                                                        *razla@post.bgu.ac.il*
*Department of Computer Science*
*Ben-Gurion University, Israel & DeepKeep, Israel*

**Moshe Sipper**                                                        *sipper@bgu.ac.il*
*Department of Computer Science*
*Ben-Gurion University, Israel*

**Reviewed on OpenReview:** *https://openreview.net/forum?id=wvLQMHtyLk*

## Abstract

Deep neural networks (DNNs) have greatly impacted numerous fields over the past decade. Yet despite exhibiting superb performance over many problems, their black-box nature still poses a significant challenge with respect to explainability. Indeed, explainable artificial intelligence (XAI) is crucial in several fields, wherein the answer alone—sans a reasoning of how said answer was derived—is of little value. This paper uncovers a troubling property of explanation methods for image-based DNNs: by making small visual changes to the input image—hardly influencing the network's output—we demonstrate how explanations may be arbitrarily manipulated through the use of evolution strategies. Our novel algorithm, AttaXAI, a model-and-data XAI-agnostic, adversarial attack on XAI algorithms, only requires access to the output logits of a classifier and to the explanation map; these weak assumptions render our approach highly useful where real-world models and data are concerned. We compare our method's performance on two benchmark datasets—CIFAR100 and ImageNet—using four different pretrained deep-learning models: VGG16-CIFAR100, VGG16-ImageNet, MobileNet-CIFAR100, and Inception-v3-ImageNet. We find that the XAI methods can be manipulated without the use of gradients or other model internals. AttaXAI successfully manipulates an image such that several XAI methods output a specific explanation map. To our knowledge, this is the first such method in a black-box setting, and we believe it has significant value where explainability is desired, required, or legally mandatory. The code is available at `https://github.com/razla/Foiling-Explanations-in-Deep-Neural-Networks`.

**Keywords:** deep learning, computer vision, adversarial attack, evolutionary algorithm, explainable artificial intelligence

## 1 Introduction

Recent research has revealed that deep learning-based, image-classification systems are vulnerable to adversarial instances, which are designed to deceive algorithms by introducing perturbations to benign images (Carlini & Wagner, 2017; Madry et al., 2017; Xu et al., 2018; Goodfellow et al., 2014; Croce & Hein, 2020). A variety of strategies have been developed to generate adversarial instances,

---

\*Equal contribution

and they fall under two broad categories, differing in the underlying threat model: white-box attacks (Moosavi-Dezfooli et al., 2016; Kurakin et al., 2018) and black-box attacks (Chen et al., 2017; Lapid et al., 2022).

In a white box attack, the attacker has access to the model's parameters, including weights, gradients, etc'. In a black-box attack, the attacker has limited information or no information at all; the attacker generates adversarial instances using either a different model, a model's raw output (also called logits), or no model at all, the goal being for the result to transfer to the target model (Tramèr et al., 2017; Inkawhich et al., 2019).

In order to render a model more interpretable, various explainable algorithms have been conceived. Van Lent et al. (2004) coined the term *Explainable Artificial Intelligence* (XAI), which refers to AI systems that "can explain their behavior either during execution or after the fact". In-depth research into XAI methods has been sparked by the success of Machine Learning (ML) systems, particularly Deep Learning (DL), in a variety of domains, and the difficulty in intuitively understanding the outputs of complex models, namely, how did a DL model arrive at a specific decision for a given input.

Explanation techniques have drawn increased interest in recent years due to their potential to reveal hidden properties of deep neural networks (Došilović et al., 2018). For safety-critical applications, interpretability is essential, and sometimes even legally required.

The importance assigned to each input feature for the overall classification result may be observed through explanation maps, which can be used to offer explanations. Such maps can be used to create defenses and detectors for adversarial attacks (Walia et al., 2022; Fidel et al., 2020; Kao et al., 2022). Figures 1 and 2 show examples of explanation maps, generated by five different methods discussed in Section 2.

In this paper, we show that these explanation maps can be transformed into any target map, using only the maps and the network's output probability vector. This is accomplished by adding a perturbation to the input that is usually unnoticeable to the human eye. This perturbation has minimal effect on the neural network's output, therefore, in addition to the classification outcome, the probability vector of all classes remains virtually identical.

**Our contribution.** As we mentioned earlier, recent studies have shown that adversarial attacks may be used to undermine DNN predictions (Goodfellow et al., 2014; Papernot et al., 2016b; Carlini & Wagner, 2017; Lapid et al., 2022). There are several more papers regarding XAI attacks, which have also shown success on manipulating XAI, but to our knowledge, they all rely on access to the neural network's gradient, which is usually not available in real-world scenarios (Ghorbani et al., 2019; Dombrowski et al., 2019).

Herein, we propose a black-box algorithm, AttaXAI, which enables manipulation of an image through a barely noticeable perturbation, without the use of any model internals, such that the explanation fits any given target explanation. Further, the robustness of the XAI techniques are tested as well. We study AttaXAI's efficiency on 2 benchmark datasets, 4 different models, and 5 different XAI methods.

The next section presents related work. Section 3 presents AttaXAI, followed by experiments and results in Section 4. In Section 5 we analyze our results, In Section 6 we make a discussion followed by concluding remarks in Section 7.

## 2 Related Work

The ability of explanation maps to detect even the smallest visual changes was shown by Ghorbani et al. (2019), where they perturbed a given image, which caused the explanatory map to change, without any specific target.

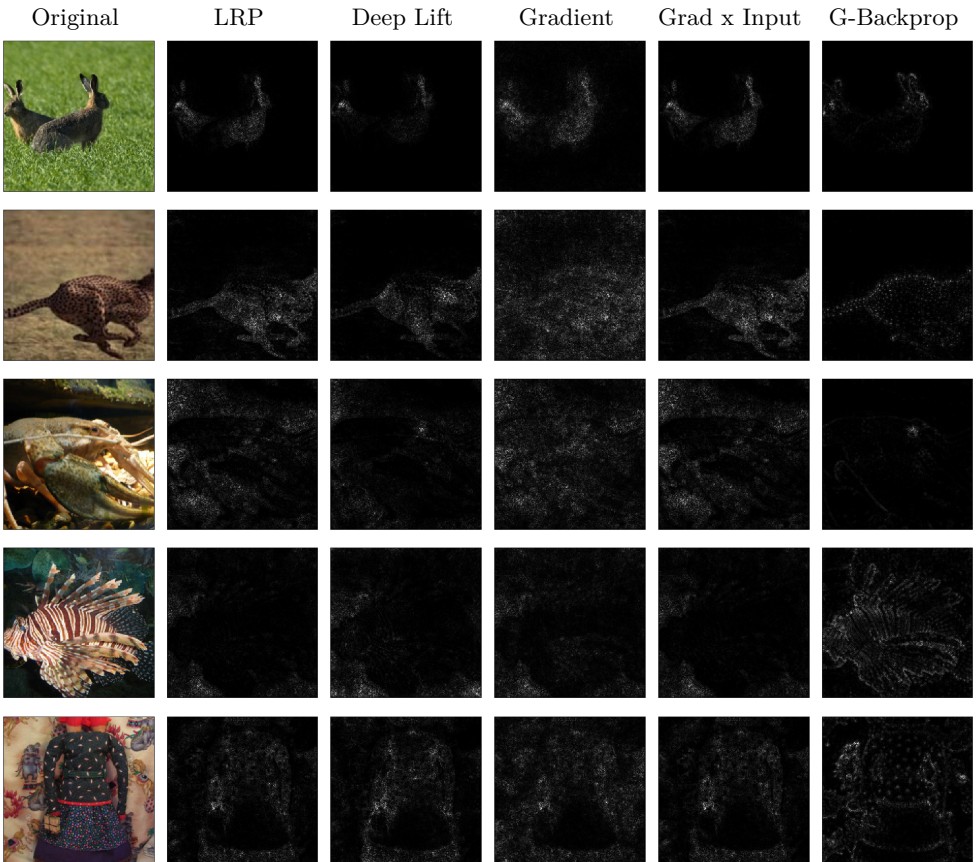

Figure 1: Explanation maps for 5 images using 5 different explanation methods. Dataset: ImageNet. Model: VGG16.

Kuppa & Le-Khac (2020) designed a black-box attack to examine the security aspects of the gradient-based XAI approach, including consistency, accuracy, and confidence, using tabular datasets.

Zhang et al. (2020) demonstrated a class of white-box attacks that provide adversarial inputs, which deceive both the interpretation models and the deep-learning models. They studied their method using four different explanation algorithms.

Xu et al. (2018) demonstrated that a subtle adversarial perturbation intended to mislead classifiers might cause a significant change in a class-specific network interpretability map.

The goal of Dombrowski et al. (2019) was to precisely replicate a given target map using gradient descent with respect to the input image. Although this work showed an intriguing phenomenon, it is a less-realistic scenario, since the attacker has full access to the targeted model.

We aimed to veer towards a more-realistic scenario and show that we can achieve similar results using no information about the model besides the probability output vector and the explanation map. To our knowledge, our work is the first to introduce a black-box attack on XAI gradient-based methods in the domain of image classification.

We will employ the following explanation techniques in this paper:

1. **Gradient**: Utilizing the saliency map, $g(x) = \frac{\partial f}{\partial x}(x)$, one may measure how small perturbations in each pixel alter the prediction of the model, $f(x)$ (Simonyan et al., 2013).

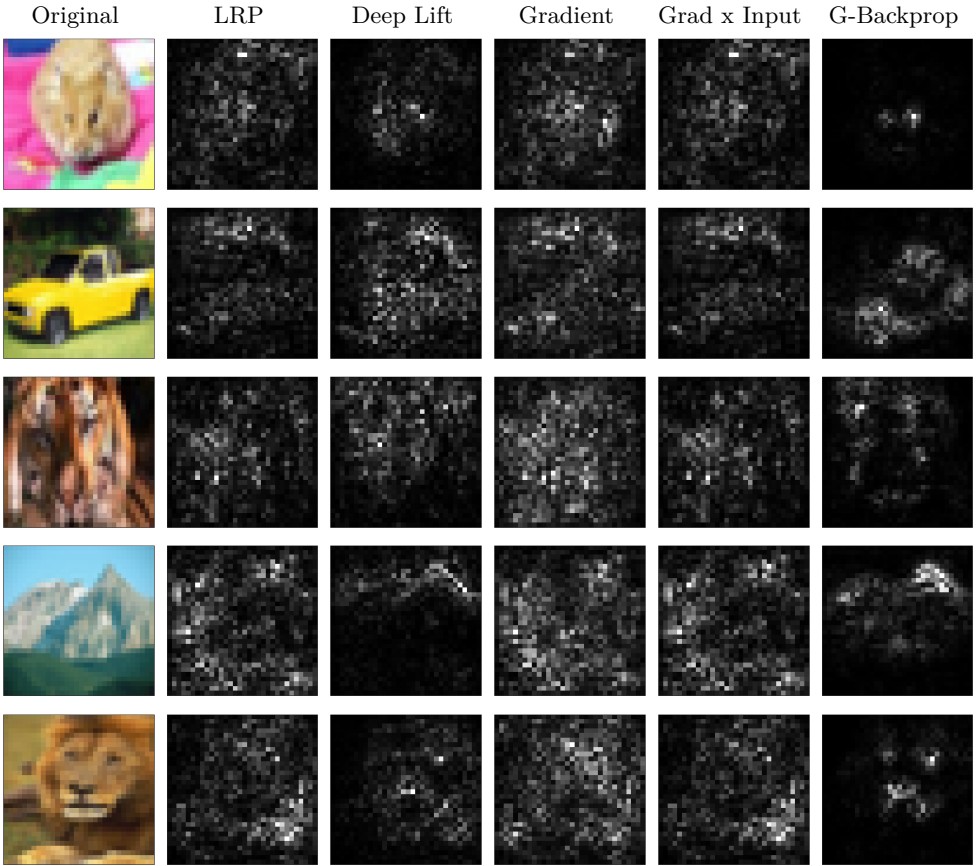

Figure 2: Explanation maps for 5 images using 5 different explanation methods. Dataset: CIFAR100. Model: VGG16.

2. **Gradient × Input**: The explanation map is calculated by multiplying the input by the partial derivatives of the output with regard to the input, $g(x) = \frac{\partial f}{\partial x}(x) \odot x$ (Shrikumar et al., 2016).

3. **Guided Backpropagation**: A variant of the Gradient explanation, where the gradient's negative components are zeroed while backpropagating through the non-linearities of the model (Springenberg et al., 2014).

4. **Layer-wise Relevance Propagation (LRP)**: With this technique, pixel importance propagates backwards across the network from top to bottom (Bach et al., 2015; Montavon et al., 2019). The general propagation rule is the following:

$$R_j = \sum_k \frac{\alpha_j \rho(w_{jk})}{\epsilon + \sum_{0,j} \alpha_j \rho(w_{jk})} R_k, \tag{1}$$

where $j$ and $k$ are two neurons of any two consecutive layers, $R_k, R_j$ are the relevance maps of layers $k$ and $j$, respectively, $\rho$ is a function that transforms the weights, and $\epsilon$ is a small positive increment.

In order to propagate relevance scores to the input layer (image pixels), the method applies an alternate propagation rule that properly handles pixel values received as input:

$$R_i = \sum_j \frac{\alpha_i w_{ij} - l_i w_{ij}^+ - h_i w_{ij}^-}{\sum_i \alpha_i w_{ij} - l_i w_{ij}^+ - h_i w_{ij}^-} R_j, \tag{2}$$

where $l_i$ and $h_i$ are the lower and upper bounds of pixel values.

5. **Deep Learning Important FeaTures (DeepLIFT)** compares a neuron's activation to its "reference activation", which then calculates contribution scores based on the difference. DeepLIFT has the potential to separately take into account positive and negative contributions, which might help to identify dependencies that other methods might have overlooked (Shrikumar et al., 2017).

## 3   AttaXAI

This section presents our algorithm, AttaXAI, discussing first evolution strategies, and then delving into the algorithmic details.

### 3.1   Evolution Strategies

Our algorithm is based on Evolution Strategies (ES), a family of heuristic search techniques that draw their inspiration from natural evolution (Beyer & Schwefel, 2002; Hansen et al., 2015). Each iteration (aka generation) involves perturbing (through mutation) a population of vectors (genotypes) and assessing their objective function value (fitness value). The population of the next generation is created by combining the vectors with the highest fitness values, a process that is repeated until a stopping condition is met.

AttaXAI belongs to the class of Natural Evolution Strategies (NES) (Wierstra et al., 2014; Glasmachers et al., 2010), which includes several algorithms in the ES class that differ in the way they represent the population, and in their mutation and recombination operators. With NES, the population is sampled from a distribution $\pi_{\psi_t}$, which evolves through multiple iterations (generations); we denote the population samples by $Z_t$. Through stochastic gradient descent NES attempts to maximize the population's average fitness, $\mathbb{E}_{Z \sim \pi_\psi}[f(Z)]$, given a fitness function, $f(\cdot)$.

A version of NES we found particularly useful for our case was used to solve common reinforcement learning (RL) problems (Salimans et al., 2017).

We used Evolution strategies (ES) in our work for the following reasons:

- **Global optima search.**   ES are effective at finding global optima in complex, high-dimensional search spaces, like those we encounter when trying to generate adversarial examples. ES can often bypass local optima that would trap traditional gradient-based methods. This ability is crucial for our work, where we assumed a black-box threat model.

- **Parallelizable computation.** ES are highly parallelizable, allowing us to to simultaneously compute multiple fitness values. This is a significant advantage given our high computational requirements; using ES allows us to significantly reduce run times.

- **No Need for backpropagation.** ES do not require gradient information and hence eschew backpropagation, which is advantageous in scenarios where the models do not have easily computable gradients or are black-box models.

**Search gradients.**   The core idea of NES is to use search gradients to update the parameters of the search distribution (Wierstra et al., 2014). The search gradient can be defined as the gradient of the expected fitness: Denoting by $\pi$ a distribution with parameters $\psi$, $\pi(z|\psi)$ is the probability density

function of a given sample $z$. With $f(z)$ denoting the fitness of a sample $z$, the expected fitness under the search distribution can be written as:

$$J(\psi) = \mathbb{E}_\psi[f(z)] = \int f(z)\pi(z|\psi)dz. \tag{3}$$

The gradient with respect to the distribution parameters can be expressed as:

$$
\begin{aligned}
\nabla_\theta J(\theta) &= \nabla_\theta \int f(z)\pi(z|\theta)dz \\
&= \int f(z)\nabla_\theta \pi(z|\theta)dz \\
&= \int f(z)\nabla_\theta \pi(z|\theta)\frac{\pi(z|\theta)}{\pi(z|\theta)}dz \\
&= \int [f(z)\nabla_\theta \log \pi(z|\theta)]\pi(z|\theta)dz \\
&= \mathbb{E}_\theta[f(z)\nabla_\theta \log \pi(z|\theta)]
\end{aligned}
$$

From these results we can approximate the gradient with Monte Carlo (Metropolis & Ulam, 1949) samples $z_1, z_2, ..., z_\lambda$:

$$\nabla_\theta J(\theta) \approx \frac{1}{\lambda}\sum_{k=1}^{\lambda} f(z_k)\nabla_\theta \log \pi(z_k|\theta) \tag{4}$$

In our experiment we sampled from a Gaussian distribution for calculating the search gradients.

**Latin Hypercube Sampling (LHS).** LHS is a form of stratified sampling scheme, which improves the coverage of the sampling space. It is done by dividing a given cumulative distribution function into $M$ non-overlapping intervals of equal $y$-axis length, and randomly choosing one value from each interval to obtain $M$ samples. It ensures that each interval contains the same number of samples, thus producing good uniformity and symmetry (Wang et al., 2022). We used both LHS and standard sampling in our experimental setup.

## 3.2 Algorithm

AttaXAI is an evolutionary algorithm (EA), which explores a space of images for adversarial instances that fool a given explanation method. This space of images is determined by a given input image, a model, and a loss function. The algorithm generates a perturbation for the given input image such that it fools the explanation method.

More formally, we consider a neural network, $f\colon \mathbb{R}^{h,w,c} \to \mathbb{R}^K$, which classifies a given image, $x \in \mathbb{R}^{h,w,c}$, where $h, w, c$ are the image's height, width, and channel count, respectively, to one of $K$ predetermined categories, with the predicted class given by $k = \arg\max_i f(x)_i$. The explanation map, which is represented by the function, $g\colon \mathbb{R}^{h,w,c} \to \mathbb{R}^{h,w}$, links each image to an explanation map, of the same height and width, where each coordinate specifies the influence of each pixel on the network's output.

AttaXAI explores the space of images through evolution, ultimately producing an adversarial image; it does so by continually updating a Gaussian probability distribution, used to sample the space of images. By continually improving this distribution the search improves.

We begin by sampling perturbations from an isotropic normal distribution $\mathcal{N}(\mathbf{0}, \sigma^2\mathbf{I})$. Then we add them to the original image, $x$, and feed them to the model. By doing so, we can approximate

the gradient of the expected fitness function. With an approximation of the gradient at hand we can advance in that direction by updating the search distribution parameters. A schematic of our algorithm is shown in Figure 3, with a full pseudocode provided in Algorithm 1.

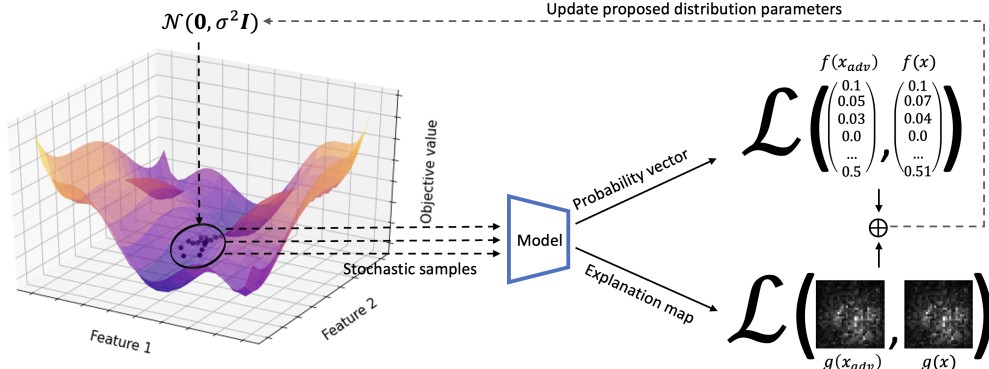

Figure 3: Schematic of proposed algorithm. Individual perturbations are sampled from the population's distribution $\mathcal{N}(\mathbf{0}, \sigma^2\mathbf{I})$, and feed into the model (Feature 1 and Feature 2 are image features, e.g., two pixel values; in reality the dimensionality is much higher). Then, the fitness function, i.e. the loss, is calculated using the output probability vectors and the explanation maps to approximate the gradient and update the distribution parameters.

### 3.3 Fitness Function

Given an image, $x \in \mathbb{R}^{h,w,c}$, a specific explanation method, $g\colon \mathbb{R}^{h,w,c} \to \mathbb{R}^{h,w}$, a target image, $x_{target}$, and a target explanation map, $g(x_{target})$, we seek an adversarial perturbation, $\delta \in \mathbb{R}^{h,w,c}$, such that the following properties of the adversarial instance, $x_{adv} = x + \delta$, hold:

1. The network's prediction remains almost constant, i.e., $f(x) \approx f(x_{adv})$.

2. The explanation vector of $x_{adv}$ is close to the target explanation map, $g(x_{target})$, i.e., $g(x_{adv}) \approx g(x_{target})$.

3. The adversarial instance, $x_{adv}$, is close to the original image, $x$, i.e., $x \approx x_{adv}$.

We achieve such perturbations by optimizing the following fitness function of the evolutionary algorithm:

$$\mathcal{L} = \alpha\|g(x_{adv}) - g(x_{target})\| + \beta\|f(x_{adv}) - f(x)\| \tag{5}$$

The first term ensures that the altered explanation map, $g(x_{adv})$, is close to the target explanation map, $g(x_{target})$; the second term pushes the network to produce the same output probability vector. The hyperparameters, $\alpha, \beta \in \mathbb{R}^{+}$, determine the respective weightings of the fitness components.

In order to use our approach, we only need the output probability vector, $f(x_{adv})$, and the target explanation map, $g(x_{target})$. Unlike white-box methods, we do not presuppose anything about the targeted model, its architecture, dataset, or training process. This makes our approach more realistic.

Minimizing the fitness value is the ultimate objective. Essentially, the value is better if the proper class's logit remains the same and the explanation map looks similar to the targeted explanation map:

$$\operatorname*{argmin}_{x_{adv}} \mathcal{L} = \alpha\|g(x_{adv}) - g(x_{target})\| + \beta\|f(x_{adv}) - f(x)\| \tag{6}$$

---

**Algorithm 1** AttaXAI

---

**Input:**

    $x \leftarrow$ original image
    $y \leftarrow$ original image's label
    $x_{expl} \leftarrow$ target explanation map
    $x_{pred} \leftarrow$ logits values of $x$
    $model \leftarrow$ model to be used
    $G \leftarrow$ maximum number of generations
    $\lambda \leftarrow$ population size
    $\sigma \leftarrow$ initial standard deviation value
    $\alpha \leftarrow$ explanation loss weight
    $\beta \leftarrow$ prediction loss weight
    $\eta_{\widehat{x}} \leftarrow$ mean learning rate
    $\eta_\sigma \leftarrow$ standard deviation learning rate

**Output:**

    $\widehat{x} \leftarrow$ adversarial image

  # Main loop
1:  $\widehat{x} \leftarrow x$
2:  **for** $g = 1, 2, ..., G$ **do**
3:     **for** $k = 1, 2, ..., \lambda$ **do**
4:         draw sample $z_k \sim \mathcal{N}(\widehat{x},\ \sigma^2 \mathbf{I})$   # $z_k$ is used to perturb an image
5:         evaluate fitness $f(z_k) = $ FITNESS$(z_k)$
6:         calculate log-derivative $\nabla_{\widehat{x}} \log \mathcal{N}(z_k | \widehat{x},\ \sigma^2)$
7:         calculate log-derivative $\nabla_\sigma \log \mathcal{N}(z_k | \widehat{x},\ \sigma^2)$
8:     $\nabla_{\widehat{x}} J = \frac{1}{\lambda} \sum_{i=1}^{\lambda} f(z_k) \nabla_{\widehat{x}} \log \mathcal{N}(z_k | \widehat{x},\ \sigma^2)$
9:     $\nabla_\sigma J = \frac{1}{\lambda} \sum_{i=1}^{\lambda} f(z_k) \nabla_\sigma \log \mathcal{N}(z_k | \widehat{x},\ \sigma^2)$
10:    $\widehat{x} = \widehat{x} + \eta_V \cdot \nabla_{\widehat{x}} J$
11:    $\sigma = \sigma + \eta_\sigma \cdot \nabla_\sigma J$

12: **function** FITNESS$(z)$
13:    $z_{expl} = XAI(z, y)$
14:    $z_{pred} = model(z)$
15:    $expl_{loss} = \|x_{expl} - z_{expl}\|_2^2$
16:    $pred_{loss} = \|x_{pred} - z_{pred}\|_2^2$
17:    return $\alpha * expl_{loss} + \beta * pred_{loss}$

---

## 4  Experiments and Results

Assessing the algorithm over a particular configuration of model, dataset, and explanation technique, involves running it over 100 pairs of randomly selected images. We used 2 datasets: CIFAR100 and ImageNet (Deng et al., 2009). For CIFAR100 we used the VGG16 (Simonyan & Zisserman, 2014) and MobileNet (Howard et al., 2017) models, and for ImageNet we used VGG16 and Inception (Szegedy et al., 2015); the models are pretrained. For ImageNet, VGG16 has an accuracy of 73.3% and Inception-v3 has an accuracy of 78.8%. For CIFAR100, VGG16 has an accuracy of 72.9% and MobileNet has an accuracy of 69.0% (these are top-1 accuracy values; for ImageNet, top5 accuracy values are: VGG16 – 91.5%, Inception-v3 – 94.4%, and for CIFAR100, VGG16 – 91.2%, MobileNet – 91.0%). We chose these models because they are commonly used in the Computer Vision community for many downstream tasks (Haque et al., 2019; Bhatia et al., 2019; Ning et al., 2017; Younis et al., 2020; Venkateswarlu et al., 2020).

The experimental setup is summarized in Algorithm 2: Choose 100 random image pairs from the given dataset. For each image pair compute a target explanation map, $g(x_{target})$, for one of the two images. With a budget of $50,000$ queries to the model, Algorithm 1 perturbs the second image, aiming to replicate the desired $g(x_{target})$. We assume the model outputs both the output probability vector and the explanation map per each query to the model—which is a realistic scenario nowadays, with XAI algorithms being part of real-world applications (Payrovnaziri et al., 2020; Giuste et al., 2022; Tjoa & Guan, 2020).

---

**Algorithm 2** Experimental setup (per dataset and model)

---

**Input:**

    $dataset \leftarrow$ dataset to be used
    $model \leftarrow$ model to be used
    $G \leftarrow$ maximum number of generations
    $\lambda \leftarrow$ population size
    $\sigma \leftarrow$ initial standard deviation value
    $\alpha \leftarrow$ explanation-loss weight
    $\beta \leftarrow$ prediction-loss weight

**Output:**

    Performance scores

1: **for** $i \leftarrow 1$ to *100* **do**
2:     Randomly choose a pair of images $x$ and $x_{target}$ from *dataset*
3:     Generate $x_{adv}$ by running Algorithm 1, with $x$ and $x_{target}$ (and all other input parameters)
4:     Save performance statistics

---

In order to balance between the two contradicting terms in Equation 5, we chose hyperparameters that empirically proved to work, in terms of forcing the optimization process to find a solution that satisfies the two objectives, following the work done in (Dombrowski et al., 2019): $\alpha = 1e11, \beta = 1e6$ for ImageNet, and $\alpha = 1e7, \beta = 1e6$ for CIFAR100. After every generation the learning rate was decreased through multiplication by a factor of 0.999. We tested drawing the population samples, both independent and identically distributed (iid) and through Latin hypercube sampling (LHS). The generation of the explanations was achieved by using the repository Captum (Kokhlikyan et al., 2020), a unified and generic model interpretability library for PyTorch.

Figures 4 through 7 shows samples of our results. Specifically, Figure 4 shows AttaXAI-generated attacks for images from ImageNet using the VGG16 model, against each of the 5 explanation methods: LRP, Deep Lift, Gradient, Gradient x Input, Guided-Backpropagation; Figure 5 shows AttaXAI-generated attacks for images from ImageNet using the Inception model; Figure 6 shows AttaXAI-generated attacks for images from CIFAR100 using the VGG16 model; and Figure 7 shows AttaXAI-generated attacks for images from CIFAR100 using the MobileNet model.

Note that our primary objective has been achieved: having generated an adversarial image ($x_{adv}$), virtually identical to the original ($x$), the explanation ($g$) of the adversarial image ($x_{adv}$) is now, incorrectly, that of the target image ($x_{target}$)—essentially, the two rightmost columns of Figures 4-7 are identical; furthermore, the class prediction remains the same, i.e., $\arg\max_i f(x)_i = \arg\max_i f(x_{adv})_i$.

## 5 Conclusions

We examined the multitude of runs in-depth, producing several graphs, which are provided in full in the extensive Appendix. We present the mean of the losses for various query budgets in Table 1.

Below, we summarize several observations we made:

- Our algorithm was successful in that $f(x_{adv}) \approx f(x)$ for all $x_{adv}$ generated, and when applying $\arg\max$ the original label remained unchanged.

- For most hyperparameter values examined, our approach converges for ImageNet using VGG, reaching $\approx 1e - 10$ MSE loss between $g(x_{adv})$ and $g(x_{target})$, for every XAI except Guided Backpropagation—which was found to be more robust then other techniques in this configuration, $\times 10$ more robust.

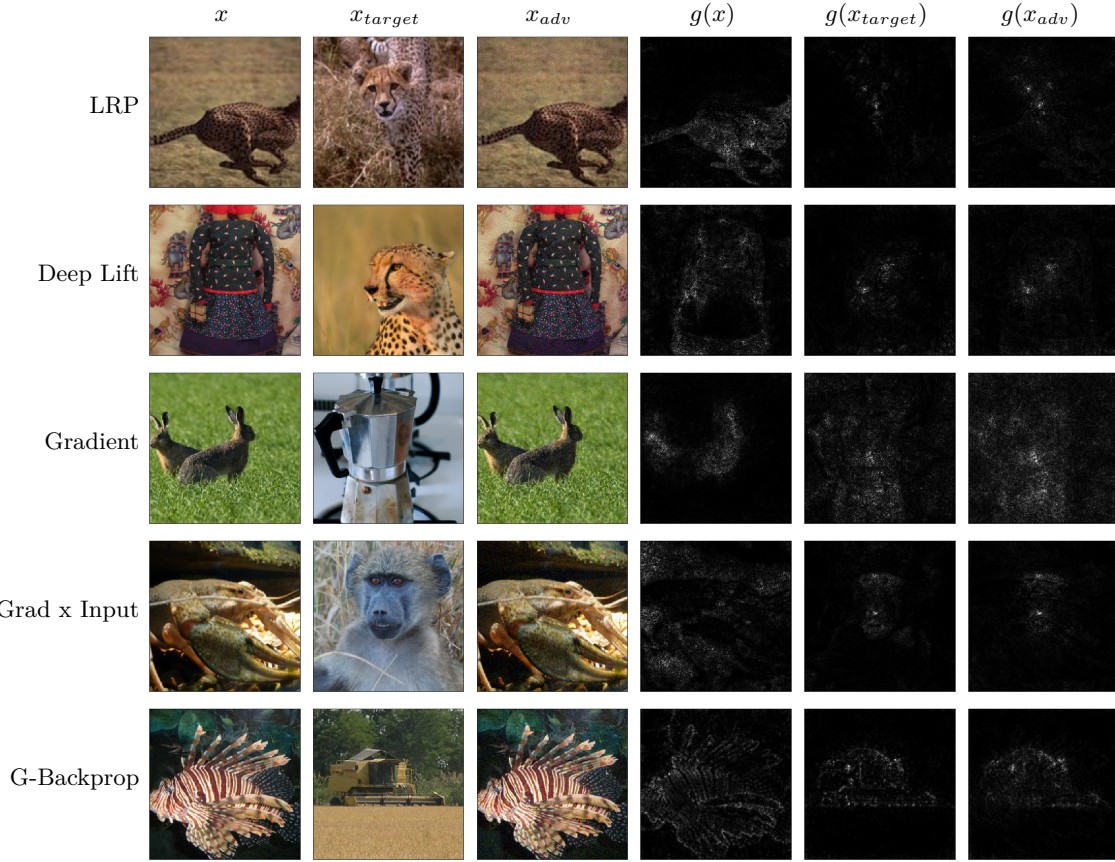

Figure 4: Attacks generated by AttaXAI. Dataset: ImageNet. Model: VGG16. Shown for the 5 explanation methods, described in the text: LRP, Deep Lift, Gradient, Gradient x Input, Guided Backpropagation (denoted G-Backprop in the figure). Note that our primary objective has been achieved: having generated an adversarial image ($x_{adv}$), virtually identical to the original ($x$), the explanation ($g$) of the adversarial image ($x_{adv}$) is now, incorrectly, that of the target image ($x_{target}$); essentially, the two rightmost columns are identical.

- For all the experiments we witnessed that the Gradient XAI method showed the smallest mean squared error (MSE) between $g(x_{adv})$ and $g(x_{target})$, i.e., it was the least robust. The larger the MSE between $g(x_{adv})$ and $g(x_{target})$ the better the explanation algorithm can handle our perturbed image.

- For VGG16 (Figures 8 and 14), Gradient XAI showed the smallest median MSE between $g(x_{adv})$ and $g(x_{target})$, while Guided Backpropagation showed the most. This means that using Gradient XAI's output as an explanation incurs the greatest risk, while using Guided Backpropagation's output as an explanation incurs the smallest risk.

- For Inception (Figure 11), Gradient XAI and Guided Backpropagation exhibited the smallest median MSE, while LRP, Gradient x Input, and Deep Lift displayed similar results.

- For the MobileNet (Figure 17), Gradient XAI exhibited the smallest MSE, while Guided Backpropagation showed the largest MSE—rendering it more robust than other techniques in this configuration.

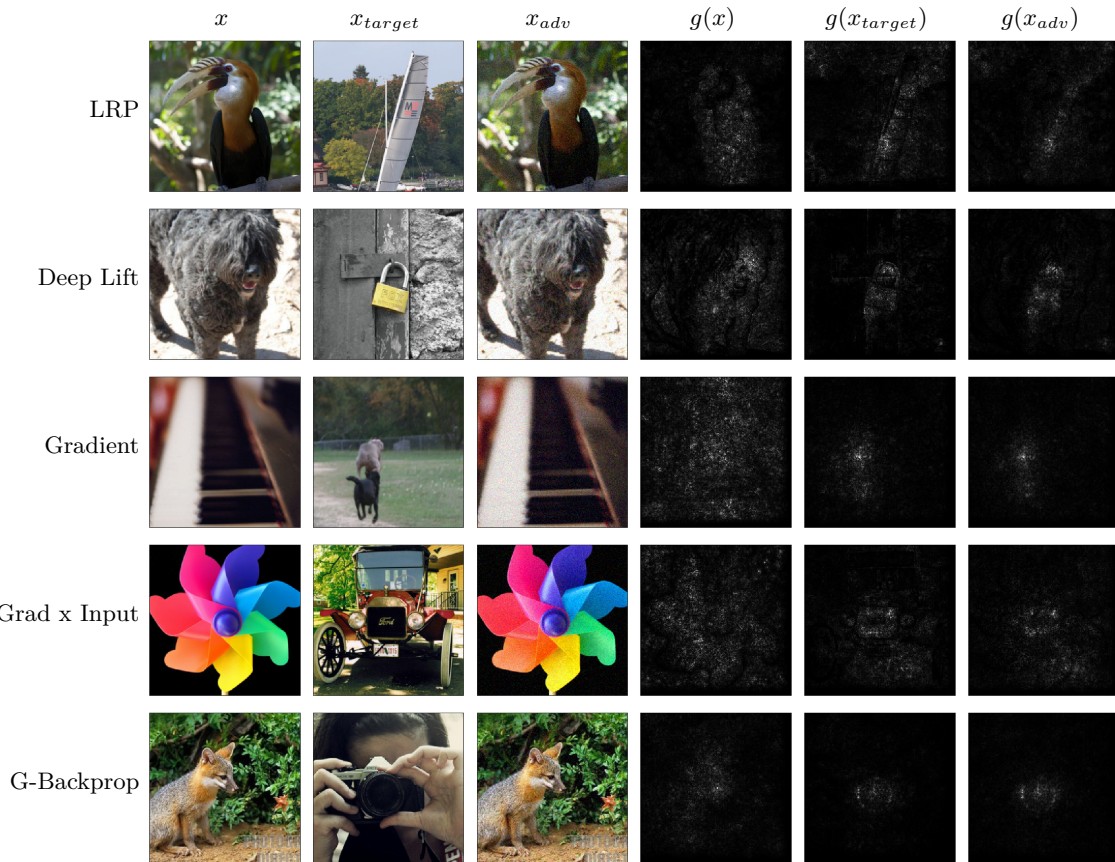

Figure 5: Attacks generated by AttaXAI. Dataset: ImageNet. Model: Inception.

Table 1: Different evaluations of our algorithm as function of number of queries to model. Best (lowest) per experiment boldfaced.

| Model | Loss | $10k$ | $20k$ | $30k$ | $40k$ | $50k$ |
|---|---|---|---|---|---|---|
| | Input | **1.0e-2** | 1.8e-2 | 2.3e-2 | 2.8e-2 | 3.4e-2 |
| VGG16-CIFAR100 | Explanation | 1.1e-6 | 9.9e-7 | 9.1e-7 | 8.6e-7 | **8.2e-7** |
| | Output | 3.6e-4 | 2.0e-4 | 1.1e-4 | 7.3e-5 | **4.1e-5** |
| | Input | **1.0e-2** | 1.7e-2 | 2.3e-2 | 2.8e-2 | 3.4e-2 |
| MobileNet-CIFAR100 | Explanation | 1.4e-6 | 1.2e-6 | 1.1e-6 | 1.1e-6 | **1.0e-6** |
| | Output | 7.4e-5 | 3.1e-5 | 1.9e-5 | 1.4e-5 | **1.3e-5** |
| | Input | **1.0e-2** | 1.7e-2 | 2.2e-2 | 2.7e-2 | 3.3e-2 |
| VGG16-ImageNet | Explanation | 1.2e-9 | 1.0e-9 | 9.0e-10 | 8.3e-10 | **7.9e-10** |
| | Output | 1.0e-5 | 7.5e-6 | 6.2e-6 | **5.4e-6** | 5.6e-6 |
| | Input | **1.0e-2** | 1.7e-2 | 2.2e-2 | 2.7e-2 | 3.3e-7 |
| Inception-v3-ImageNet | Explanation | 7.5e-10 | 6.5e-10 | 6.1e-10 | 5.8e-10 | **5.6e-10** |
| | Output | 1.9e-5 | 1.3e-5 | 1.1e-5 | **9.5e-6** | 1.0e-5 |

- MobileNet is more robust than VGG16 in that it attains higher MSE scores irrespective of the XAI method used. We surmise that this is due to the larger number of parameters in VGG16.

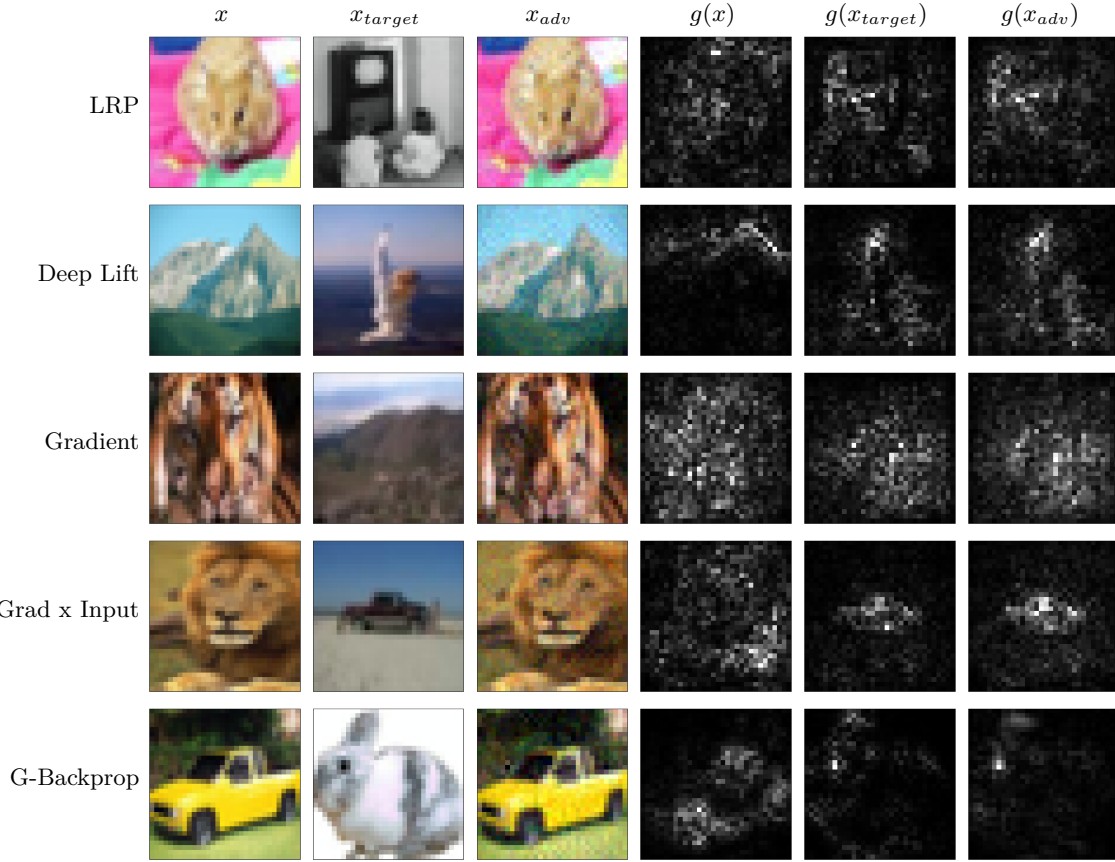

Figure 6: Attacks generated by AttaXAI. Dataset: CIFAR100. Model: VGG16.

- The query budget was 50,000 for all experiments. In many runs the distance between the target explanation and the adversarial explanation reached a plateau after roughly 25,000 queries.

## 6  Discussion

**Broader impact.** Our research aims to evidentiate vulnerabilities of explainable artificial intelligence (XAI) in adversarial attacks—which is of paramount importance for a wide array of domains.

For example, in healthcare, XAI is increasingly used in decision-making processes, such as disease diagnosis and treatment recommendations (Zhang et al., 2022; Muneer & Rasool, 2022). If these systems are susceptible to adversarial attacks, the consequences could be potentially life-threatening.

In the financial sector XAI systems are used for tasks such as fraud detection and credit-risk assessment (Cirqueira et al., 2021; Psychoula et al., 2021). Vulnerabilities could lead to substantial financial losses.

In cybersecurity, XAI is used to detect anomalies and potential threats, and weaknesses could undermine the entire security infrastructure of an organization (Srivastava et al., 2022; Capuano et al., 2022).

Thus, we believe our research has broad implications, as it aims to fost the development of robust, secure XAI systems, which retain reliability under adversarial conditions.

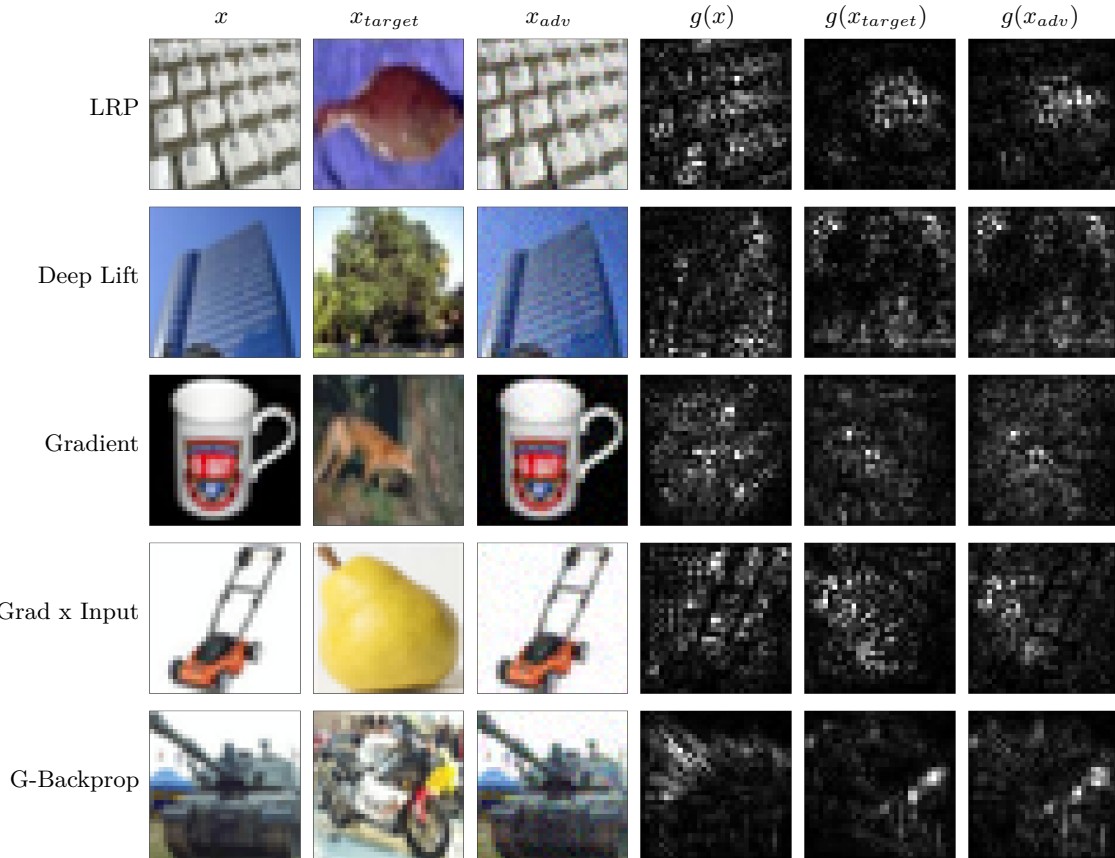

Figure 7: Attacks generated by AttaXAI. Dataset: CIFAR100. Model: MobileNet.

**Potential real-world value.** Our study offers several real-world benefits. First, by identifying the potential weaknesses in XAI systems, we provide crucial insights for organizations to take necessary precautions and develop countermeasures against adversarial attacks. This can help prevent extensive damage, both in terms of financial losses and trust erosion, which may result from a successful attack.

Additionally, our research might prompt organizations to invest more in AI security, thereby promoting the development of more resilient XAI systems. Policymakers can also utilize our research findings to create or amend legislation surrounding AI and cybersecurity. As AI technologies become increasingly integrated into our daily lives, establishing proper regulatory frameworks is crucial to ensure their safe, ethical, and responsible use. Our research provides empirical evidence that can guide these legislative efforts, contributing to a more-secure digital society.

**Query cost and adversarial effectiveness.** The relationship between query cost and adversarial effectiveness is critical in assessing attacks against deep neural networks, particularly for manipulating XAI maps. This paper's proposed black-box attack leverages a query budget of $50,000$, striking a balance between computational demand and attack success. While increasing the query budget may offer more precision, this would be resource-intensive. Conversely, a lower query budget may be less effective but more computationally efficient. The use of $50,000$ queries in our attack demonstrates a vulnerability in deep learning models that can be exploited without exorbitant computational costs, revealing potential security issues and improvement areas. Future research could focus on techniques to lower query cost while maintaining adversarial effectiveness, as well as methods to bolster model resilience against such attacks. This study, therefore, offers valuable insights into the trade-off between query cost and adversarial effectiveness.

**Future directions.** Our research also paves the way for future studies on adversarial attacks. For one, while our study focuses on certain types of XAI techniques, future research could explore the susceptibility of other XAI techniques to adversarial attacks, broadening our understanding of this critical security issue.

Further, we highlight the need for additional research on the development of more-advanced defensive mechanisms against adversarial attacks. As adversaries continually evolve their tactics, it is imperative that our defenses evolve as well.

Moreover, our research underscores the importance of considering the socio-ethical implications of adversarial attacks. As AI becomes more pervasive, attacks against these systems have the potential to cause widespread disruption and harm. Thus, future research should also consider the societal and ethical consequences of these attacks, informing policies and practices aimed at mitigating these potential harms.

**In summary,** the broader impact and potential real-world values of our study are substantial, extending across various domains, ranging from healthcare to finance, to cybersecurity. By offering a foundation for future research and by suggesting practical strategies to mitigate adversarial attacks, our study contributes to the ongoing effort to secure AI systems and ensure their responsible use.

## 7 Concluding Remarks

Recently, practitioners have started to use explanation approaches more frequently. We demonstrated how focused, undetectable modifications to the input data can result in arbitrary and significant adjustments to the explanation map. We showed that explanation maps of several known explanation algorithms may be modified at will. Importantly, this is feasible with a black-box approach, while maintaining the output of the model. We tested AttaXAI against the ImageNet and CIFAR100 datasets using 4 different network models.

It is obvious that neural networks operate quite differently from humans, capturing fundamentally distinct properties. In addition, further work is required in the XAI domain to make XAI algorithms that are more reliable.

This work has shown that explanations are easily foiled—without any recourse to internal information—raising questions regarding XAI-based defenses and detectors.

This work has also investigated the robustness of various XAI methods, revealing that Gradient XAI is the least robust XAI method and Guided Backpropagation is the most robust one.

**Future suggestions.** In our study we examined how to attack a model's (XAI) explanation for a given input, prediction, and XAI method. Some questions still remain:

- A way to predict whether a XAI attack will be successful, and how many queries will be needed.

- A better metric for a successful XAI attack, since in our results we observed that a smaller L2 distance does not necessarily translate to a more "convincing" attack.

- Find a way to eliminate the need for model feedback, i.e., go fully black-box. Applying XAI attacks via transferability (Papernot et al., 2016a; Xie et al., 2019; Wang et al., 2021) might be a way to move forward.

- When developing new XAI methods find ways to render them more robust to adversarial attacks.

## Acknowledgement

We thank the reviewers for a fruitful discussion. This research was partially supported by the Israeli Innovation Authority through the Trust.AI consortium.

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

## Appendix

The following figures provide our full qualitative results. Hyperparameters: $n\_pop$ – population size of evolutionary algorithm; $lr$ – learning rate of gradient approximation step for updating the attack; $LS$ – use of Latin sampling or regular sampling.

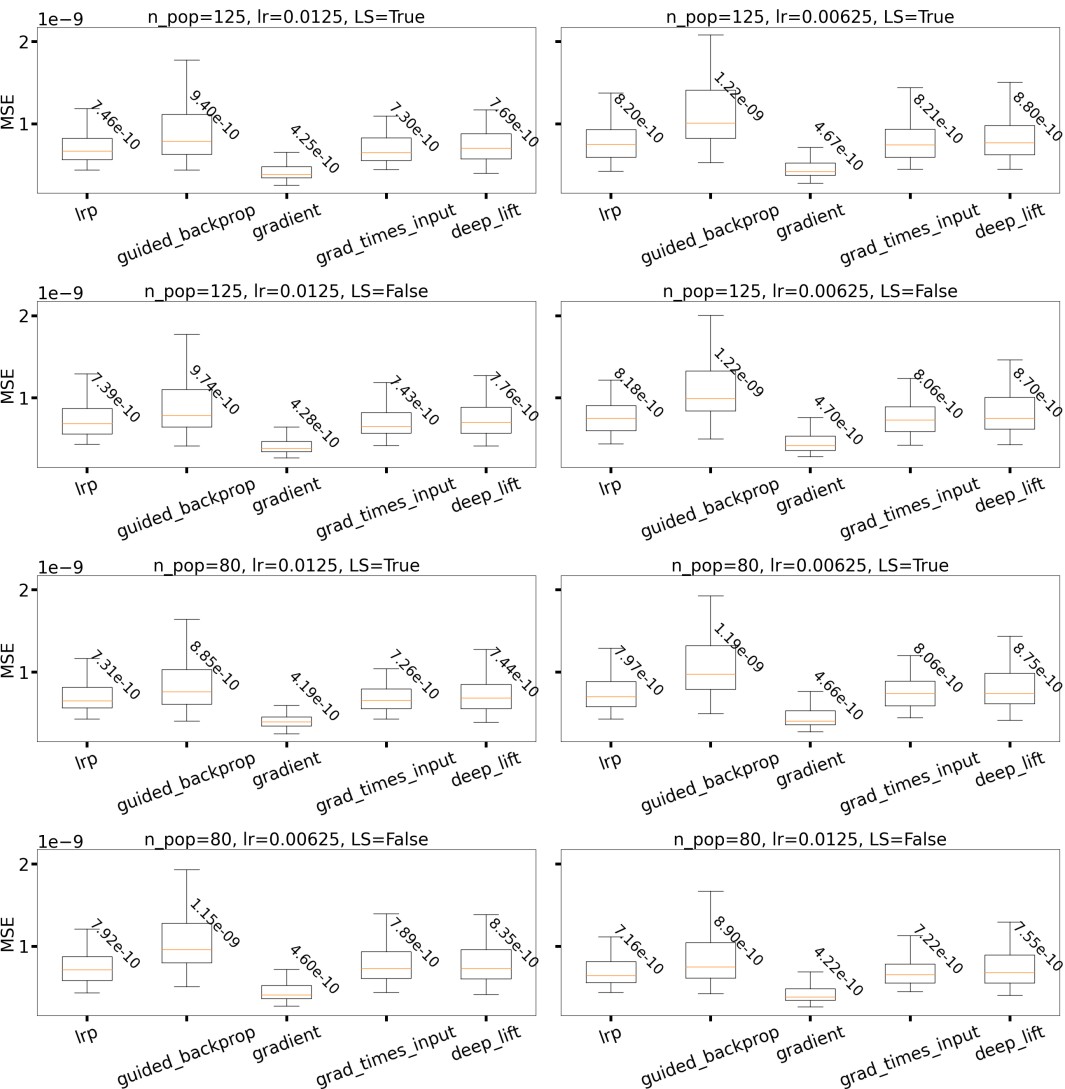

Figure 8: Similarity in terms of MSE between target explanation map, $g(x_{target})$, and best final adversarial explanation map, $g(x_{adv})$, for 8 different hyperparameter configurations. Dataset: ImageNet. Model: VGG16. The Gradient XAI method is the most susceptible to attacks while Guided backpropogation is the hardest to attack; Deep Lift, LRP, and Gradient x Input are similar.

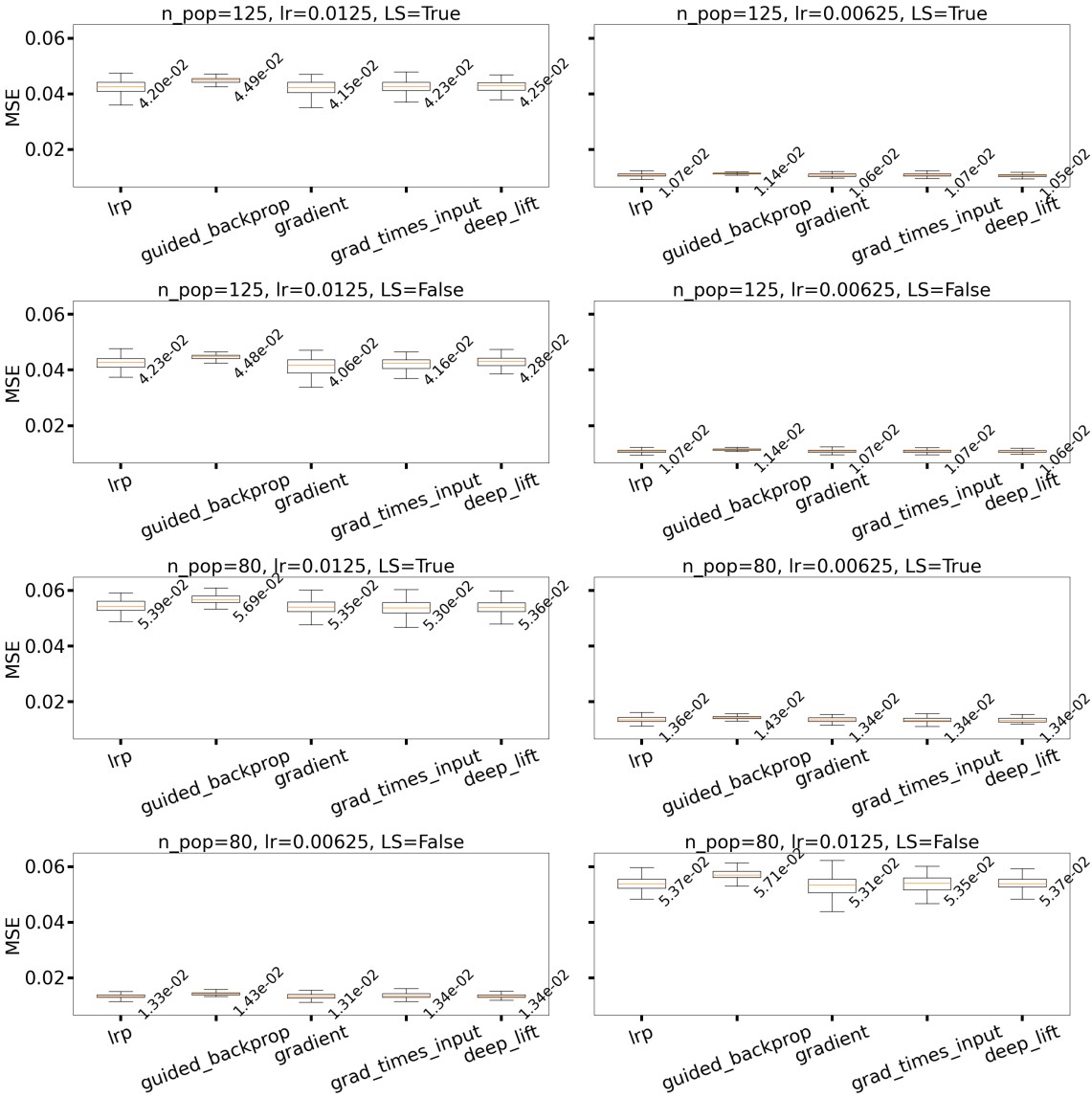

Figure 9: Similarity in terms of MSE between input image, $x$, and best final adversarial, $x_{adv}$, for 8 different hyperparameter configurations. Dataset: ImageNet. Model: VGG16. A higher learning rate and a smaller population size (i.e., more gradient steps) contribute to the perturbation of the image. The sampling method has no effect.

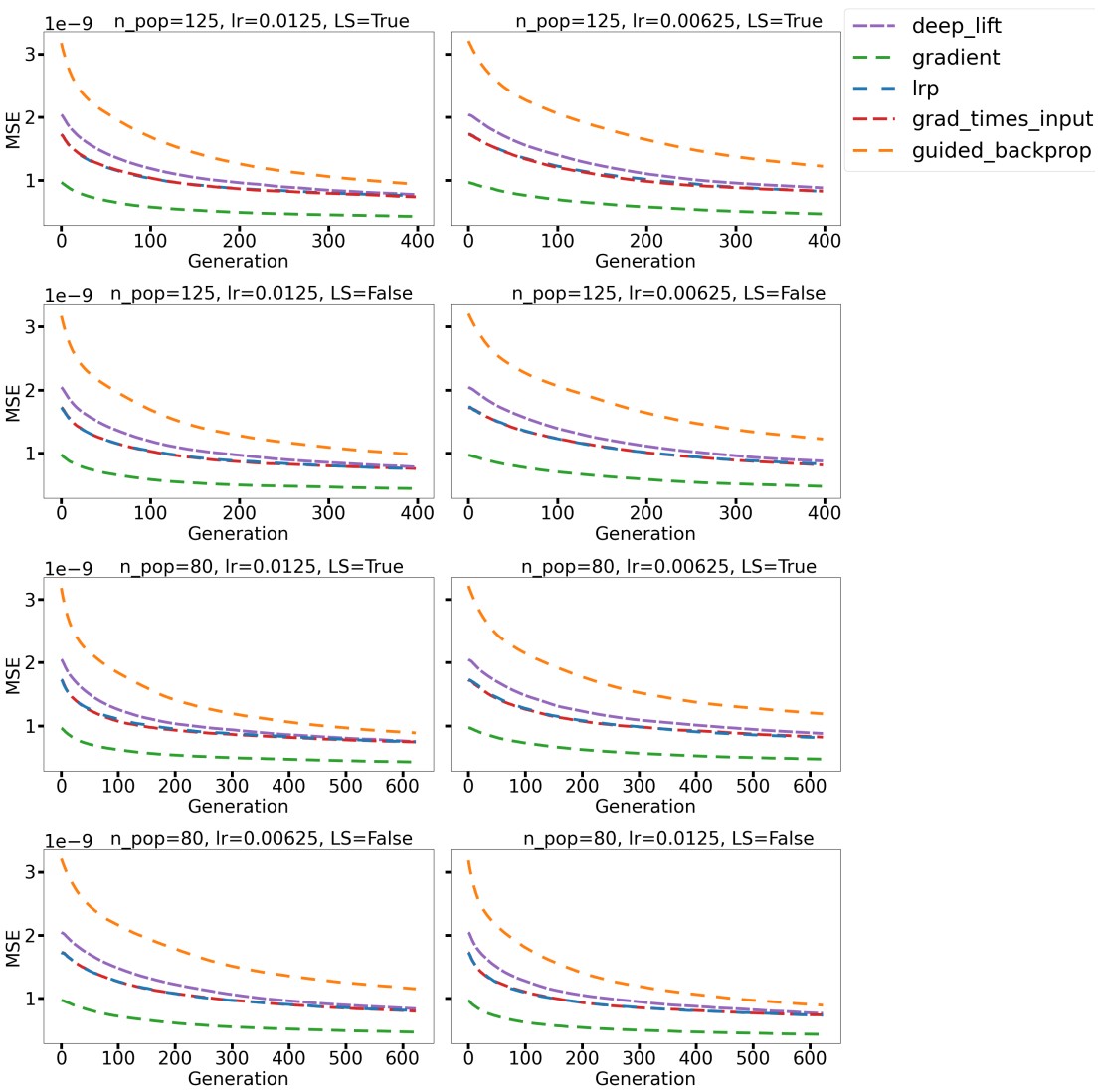

Figure 10: MSE loss value as function of evolutionary generation for 8 different hyperparameter configurations. Dataset: ImageNet. Model: VGG16.

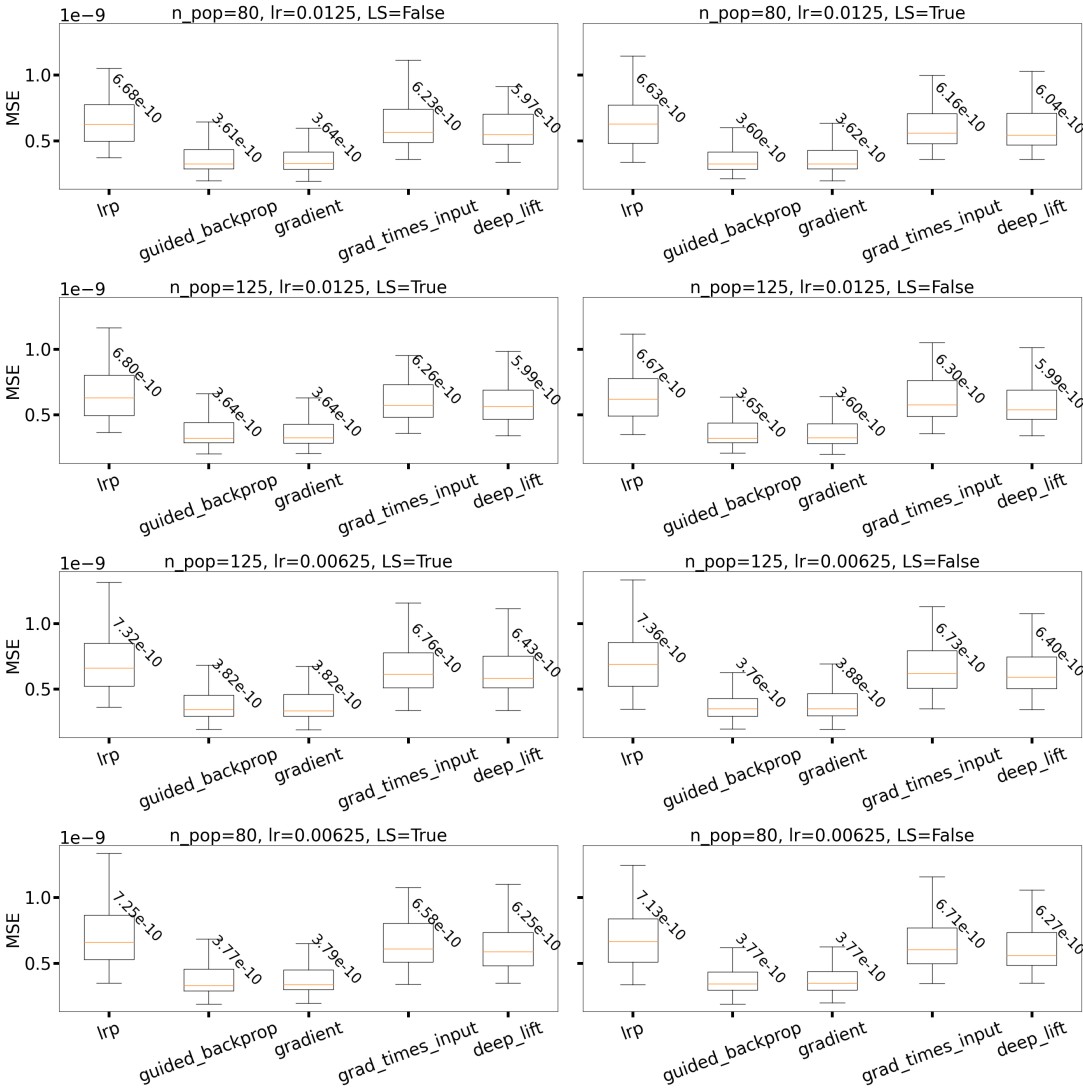

Figure 11: Similarity in terms of MSE between target explanation map, $g(x_{target})$, and best final adversarial explanation map, $g(x_{adv})$, for 8 different hyperparameter configurations. Dataset: ImageNet. Model: Inception. The Gradient XAI and Guided backpropogation methods are the most susceptible to attacks while Deep Lift, LRP, and Gradient x Input are less susceptible.

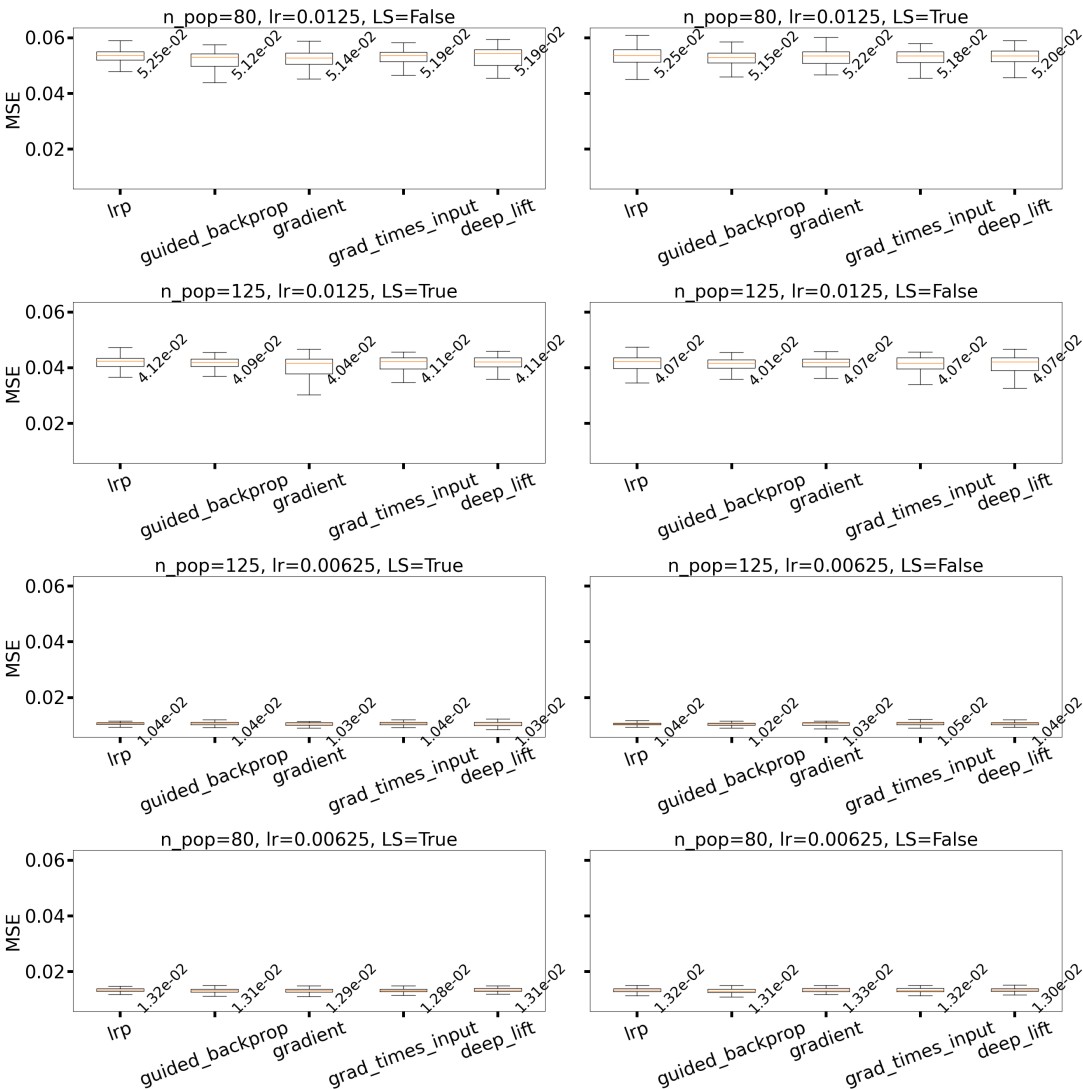

Figure 12: MSE loss value for input image versus chosen adversarial image for 8 different hyperparameter configurations. Dataset: ImageNet. Model: Inception. A higher learning rate and a smaller population size (i.e., more gradient steps) contribute to the perturbation of the image. The sampling method has no effect.

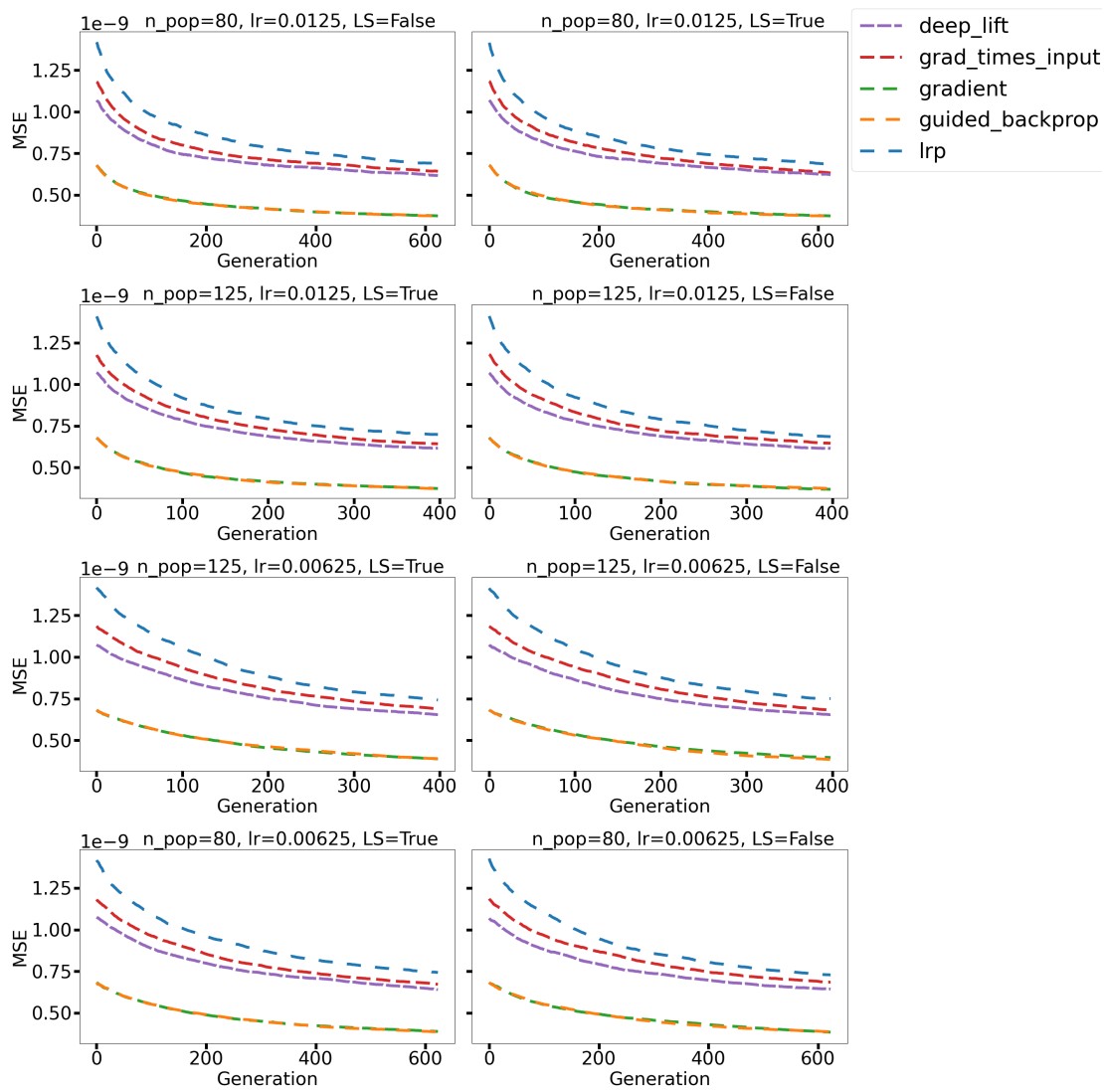

Figure 13: MSE loss value as function of evolutionary generation for 8 different hyperparameter configurations. Dataset: ImageNet. Model: Inception.

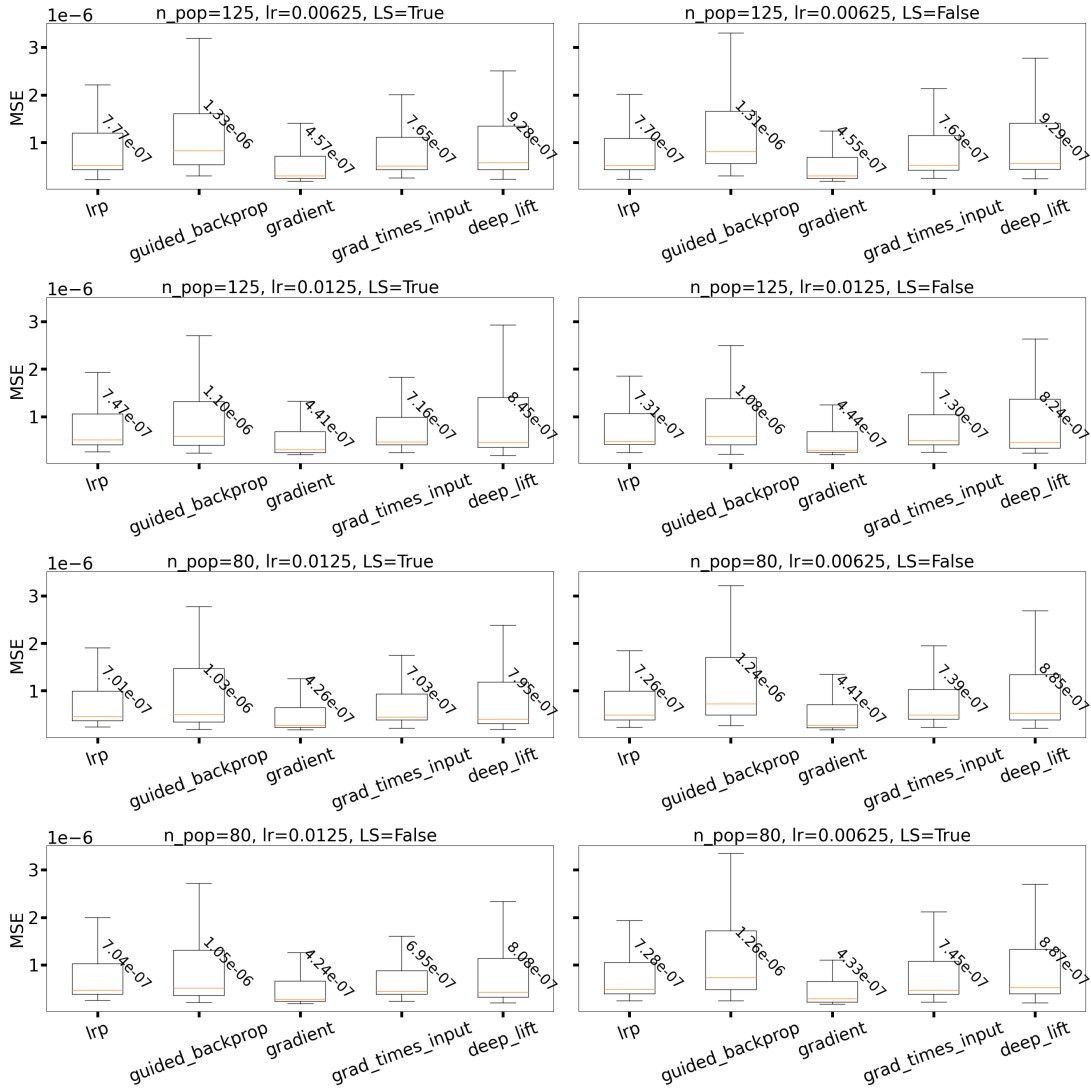

Figure 14: Similarity in terms of MSE between target explanation map, $g(x_{target})$, and best final adversarial explanation map, $g(x_{adv})$, for 8 different hyperparameter configurations. Dataset: CIFAR100. Model: VGG16. The Gradient XAI method is the most susceptible to attacks while Guided backpropogation and Deep Lift are the hardest to attack; LRP and Gradient x Input are similar.

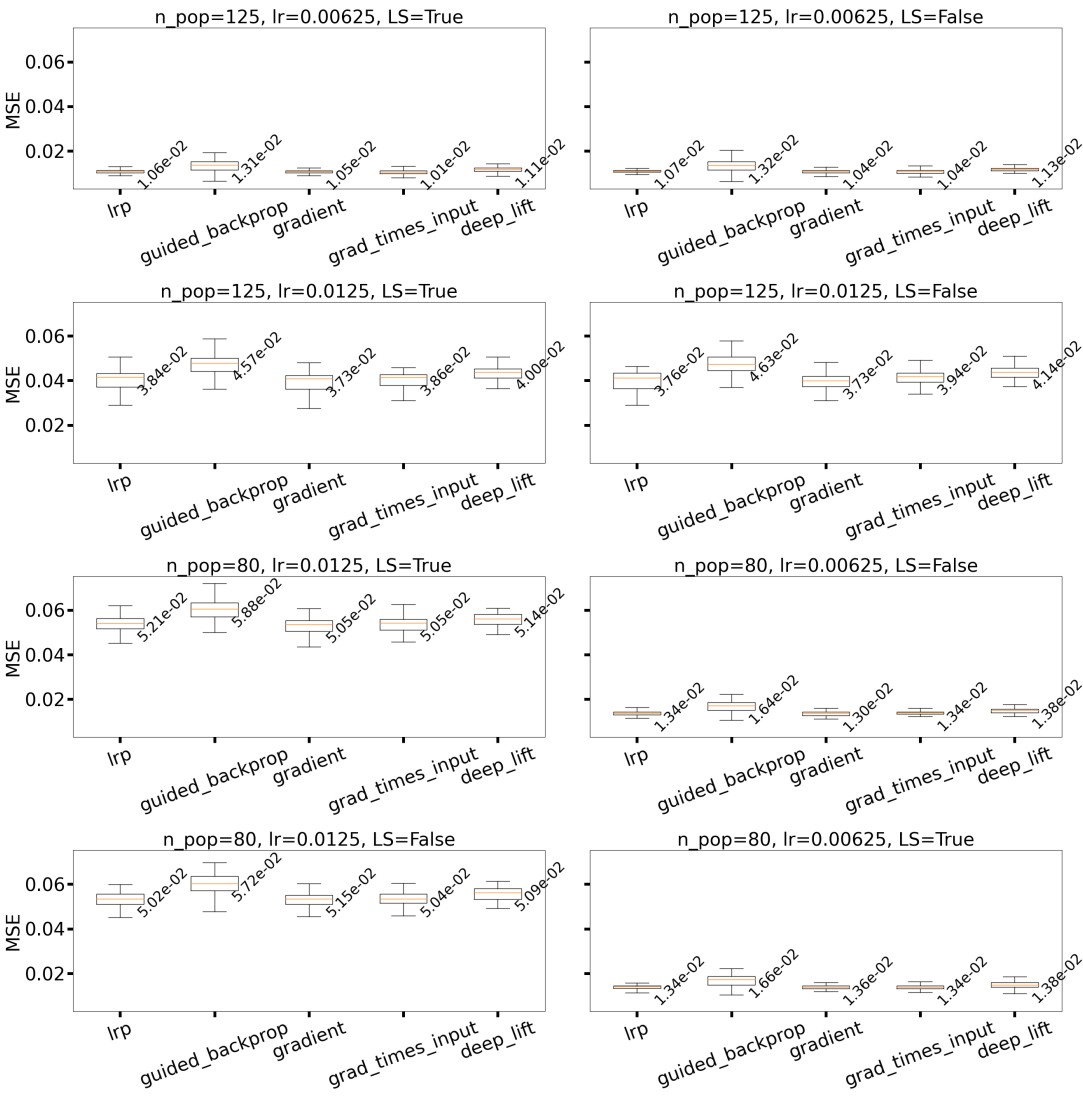

Figure 15: MSE loss value for input image versus chosen adversarial image for 8 different hyperparameter configurations. Dataset: CIFAR100. Model: VGG16. A higher learning rate and a smaller population size (i.e., more gradient steps) contribute to the perturbation of the image. The sampling method has no effect.

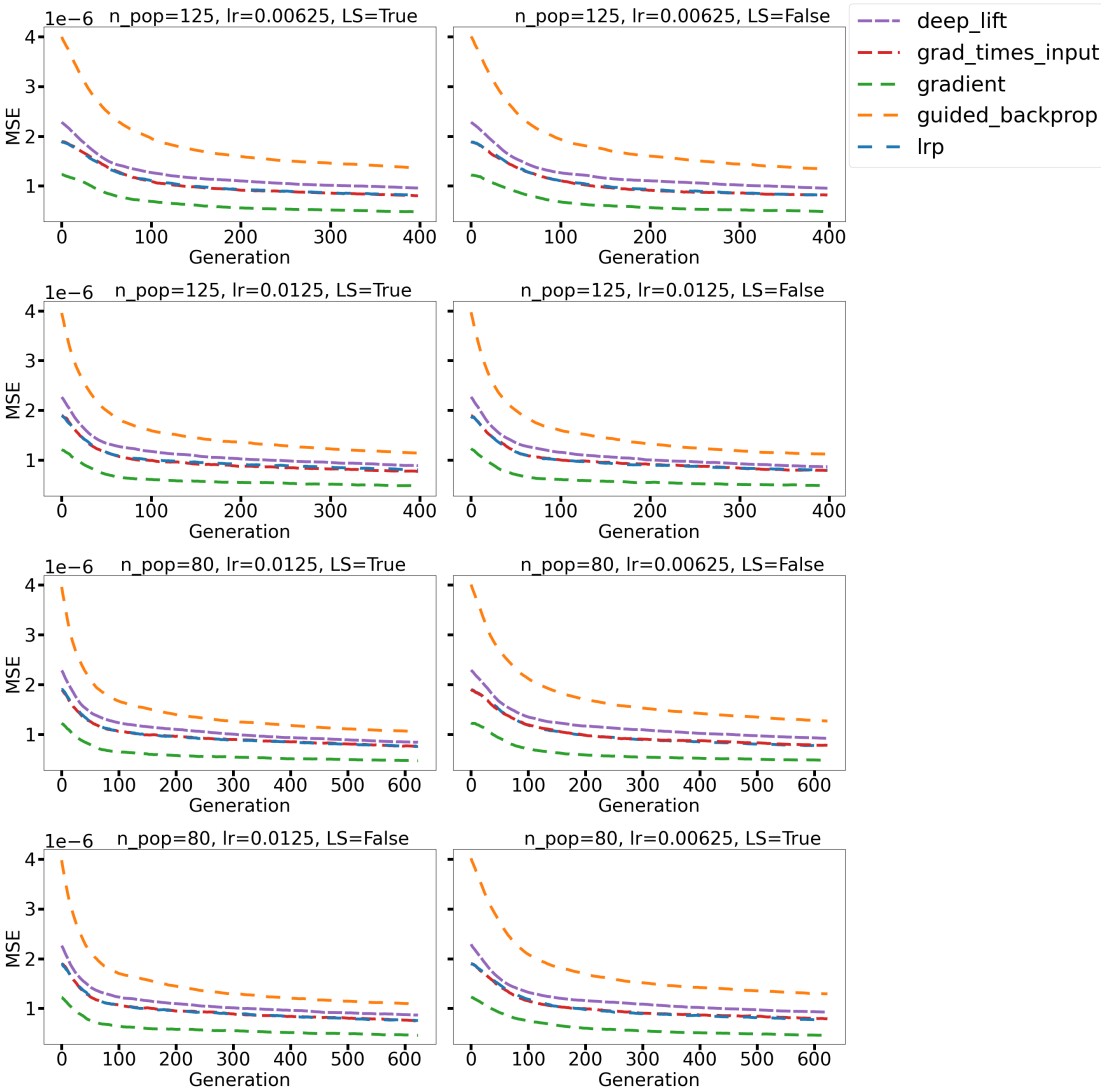

Figure 16: MSE loss value as function of evolutionary generation for 8 different hyperparameter configurations. Dataset: CIFAR100. Model: VGG16.

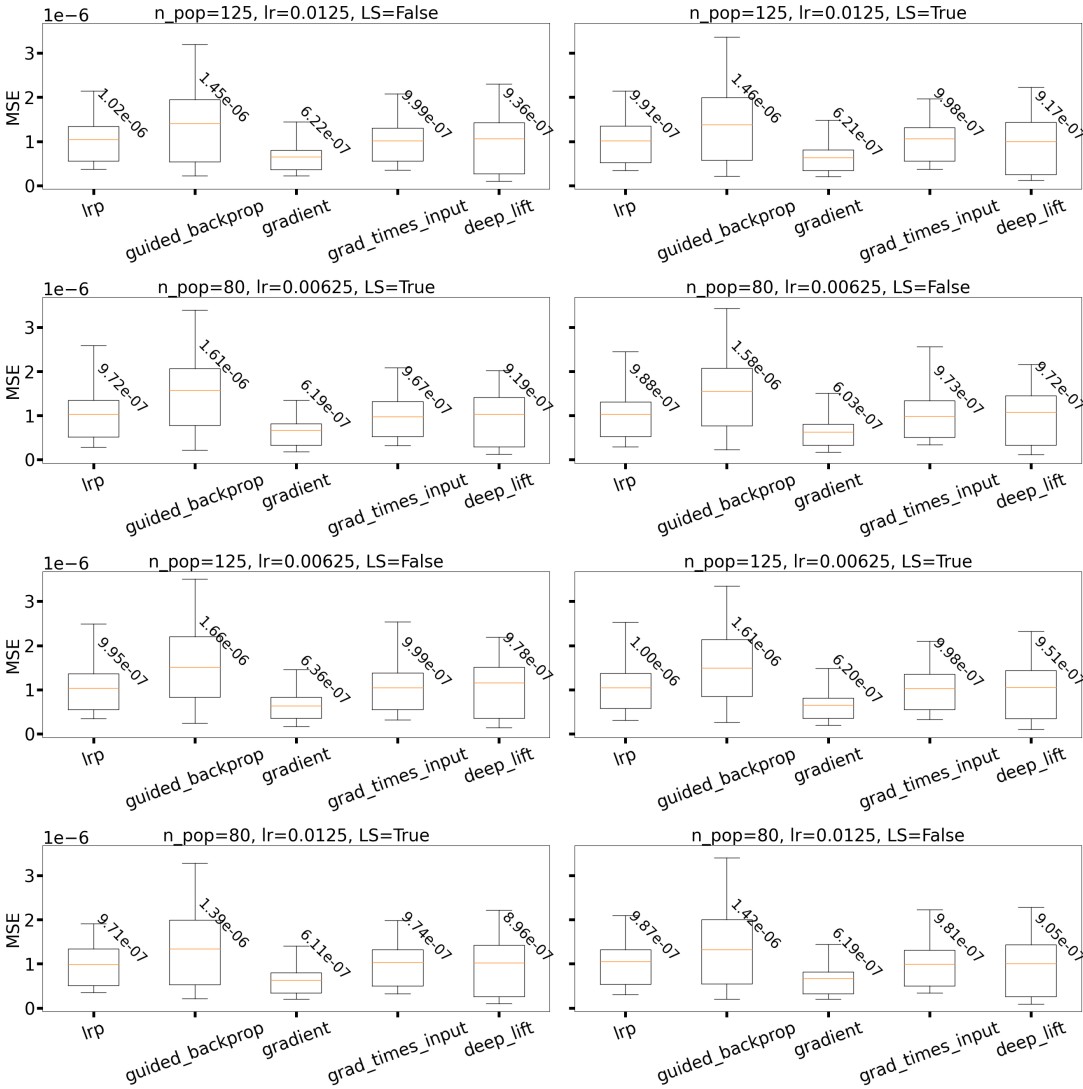

Figure 17: Similarity in terms of MSE between target explanation map, $g(x_{target})$, and best final adversarial explanation map, $g(x_{adv})$, for 8 different hyperparameter configurations. Dataset: CIFAR100. Model: MobileNet. The Gradient XAI method is the most susceptible to attacks while Guided backpropogation is the hardest to attack; Deep Lift, LRP, and Gradient x Input are similar.

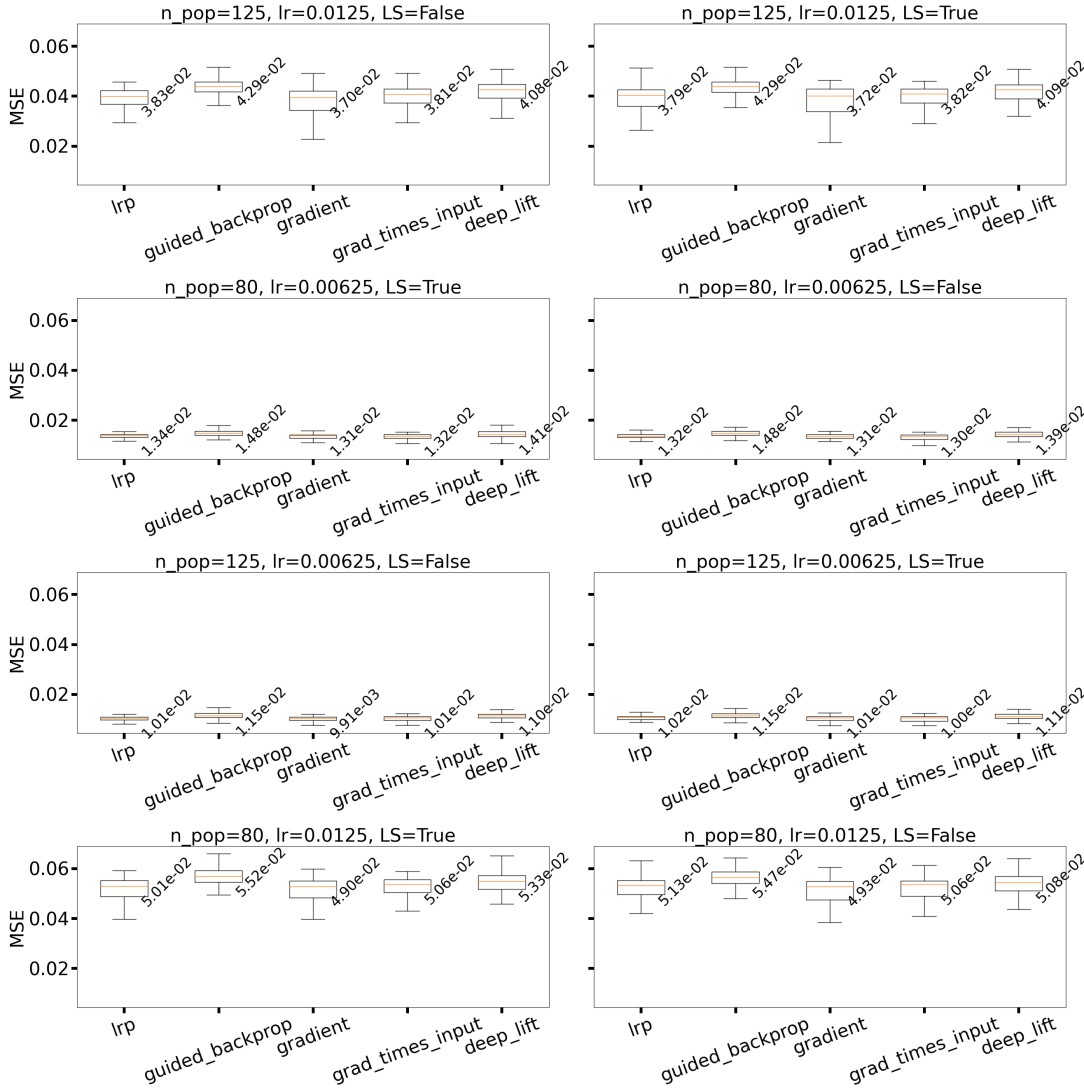

Figure 18: MSE loss value for input image versus chosen adversarial image for 8 different hyperparameter configurations. Dataset: CIFAR100. Model: MobileNet. A higher learning rate and a smaller population size (i.e., more gradient steps) contribute to the perturbation of the image. The sampling method has no effect.

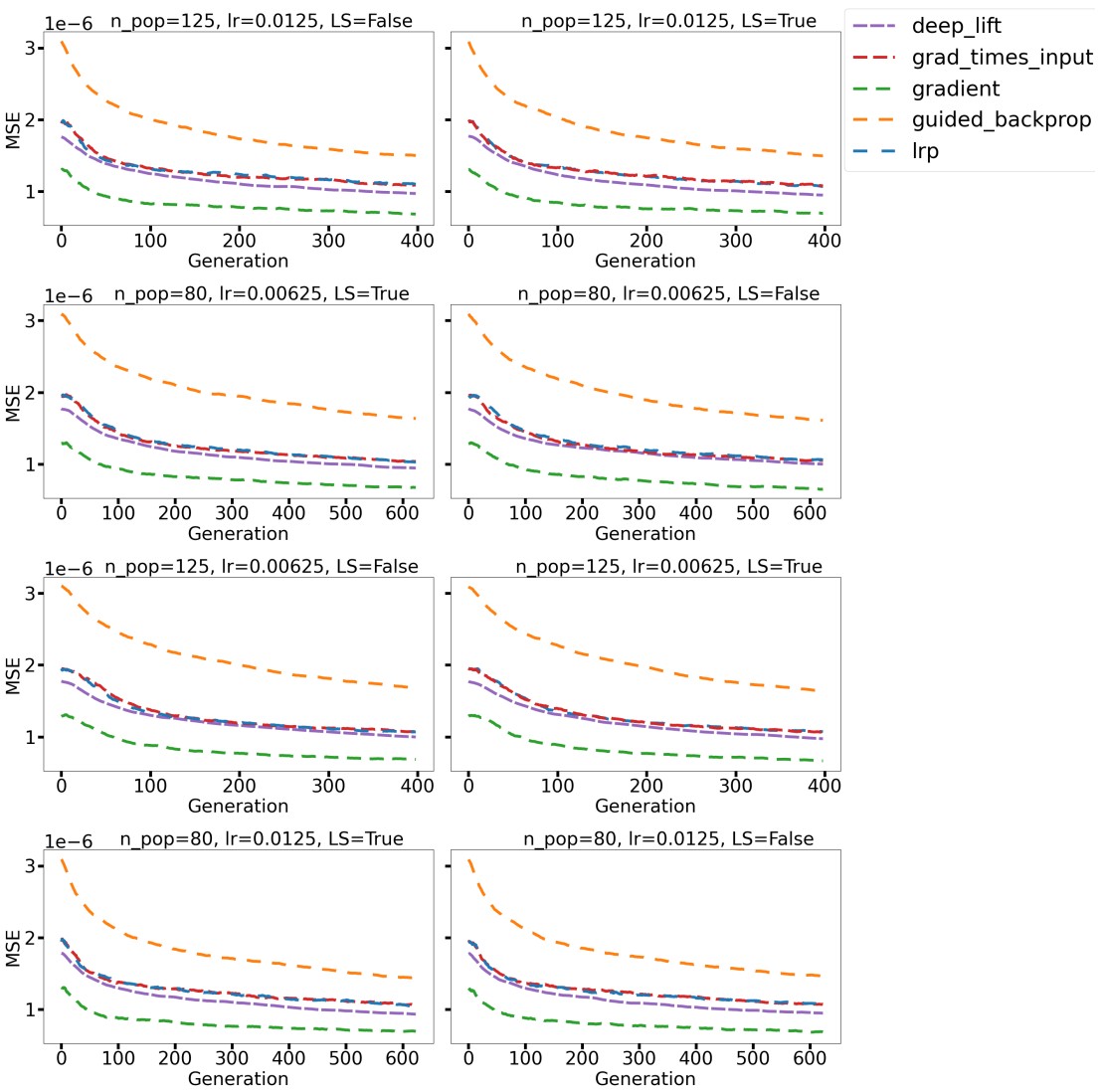

Figure 19: MSE loss value as function of evolutionary generation for 8 different hyperparameter configurations. Dataset: CIFAR100. Model: MobileNet.

