# OpenReview forum: "Foiling Explanations in Deep Neural Networks"
_TMLR — Accepted by TMLR_

### Review · Reviewer_rAzr · 2023-06-16

**Summary Of Contributions:**

The paper presents a new framework, AttaXAI, to add pixel-space perturbation on the input image (which maintains a consistent classifier output) to mutate explanation maps for DNN classifiers. The algorithm uses Natural Evolution Strategies (NES) which is model-agnostic: only requires access to (1) the model output logits and (2) the explanation map. The authors tested AttaXAI against the ImageNet and CIFAR100 datasets to show the effectiveness of the framework.

**Audience:**

Yes

**Broader Impact Concerns:**

Given that the pixel-space perturbation is hard to apply to real-world images, the authors should discuss how the proposed attack will have potential real-world applications.

**Claims And Evidence:**

Yes

**Requested Changes:**

Please address my concerns stated in the weakness section.

**Strengths And Weaknesses:**

**Strengths**
1. The algorithm is model-agnostic and data-agnostic, making it practical in real-world scenarios.
2. The authors show the effectiveness of the framework on two benchmark datasets using four different pretrained deep-learning models.
3. The paper is well-written and provides a clear storyline that motivates the proposed approach.

**Weaknesses**
1. The success of the black-box attack on explanation maps will require a sufficient number of queries to generate effective adversaries. It will be constructive if the paper elaborates more details on the query cost (e.g., time of inference/API call) and discuss how adversarial effectiveness is affected by the number of queries.
2. Authors choose to use NES for the black-box optimization. It will be validating the framework if authors can do ablation study on different types of black-box optimization.
3. The paper shall provide more details on how the crafted loss can balance two objectives: maintaining the model prediction while manipulating the explanation map.
4. Authors should analyze the perturbation budget (i.e., the numerical range of noise) imposed on the image. A too large budget will affect the image quality and make it hard to pass user observation. The paper should evaluate how the perturbation affects the image quality and perform analysis on the relation of perturbation budget with attack success rate.

---

### Review · Reviewer_Yi5a · 2023-06-21

**Summary Of Contributions:**

The work proposed an adversarial attach against explainable AI algorithm based on evolution strategy algorithm and Monte Carlo sampling to estimate the gradient of a fitness function. The attach method only relies on the classification logits and the explanation map. It demonstrates good results on a wide range of experiment settings, across different models (VGG, MobileNet) and different datasets.

**Audience:**

Yes

**Broader Impact Concerns:**

Since the work studies adversarial attacks on AI models, the paper should discuss what real-world threats the proposed approach could pose for existing XAI approaches and the values of their study in terms of dealing with such threat.

**Claims And Evidence:**

Yes

**Requested Changes:**

The paper should include discussions on broader impact and potential real-world values of its study on adversarial attacks against XAIs, which could help better motivate their study.


**Strengths And Weaknesses:**

Strengths:

1. The approach is a black-box attack and does not rely on model parameters.

2. The presentation of the approach is detailed and well-structured.

Weakness:

1. The paper does not have discussions on broader impact and potential real-world values of its study on adversarial attacks against XAIs.

2. There's no quantitative evaluation of their approach in the main paper.

---

### Review · Reviewer_PXqs · 2023-06-25

**Summary Of Contributions:**

This paper presents a method for producing image perturbations which change the output of saliency maps towards arbitrary values. The approach is evaluated on several interpetability methods and image datasets.

**Audience:**

Yes

**Claims And Evidence:**

No

**Requested Changes:**

1. Please update the claims about imperceptible perturbations to more accurately reflect the (frequently clearly perceptible) examples shown in the paper.

2. The authors should include a comprehensive summary of their quantitative results on the success rate of their method to obtain an approximation of a given explanation at a given budget for various degrees of approximation.

3. The choice of method would benefit from further justification as to why this particular form of population based search is needed.


Errata: please make appropriate use of \citep and \citet. The sudden appearance of names outside of the citation's parentheses is confusing when the citation is used in passing.


**Strengths And Weaknesses:**

Strengths:

  - The idea of constructing adversarial examples with the goal of foiling not a classifier but an explanation is an intriguing one. As far as I am aware most prior work on this topic had focused on leveraging adversarial examples with respect to to an image's classification to provide insight into the interpretability of the model. The construction of adversarial examples with respect to the explanation seems to suggest that these methods might not be as robust as previously thought, which is certainly of interest to the broader community.

  - The paper provides several nice visualizations of the adversarial examples found, which gives qualitative insight into the method's strengths and limitations, and covers several different methods of generating model explanations.

Weaknesses:

  - The adversarial examples found are visually perceptible perturbations in several instances, in contrast to the paper's claims that the perturbations are imperceptible. The perturbed photos look noticeably more grainy and often have odd patterns or small patches of colour. This suggests the method may not be finding 'true' adversarial examples in many cases, as it requires too large of a perturbation budget in order to attain the desired explanation output.

  - The main paper lacks quantitative results. How often is the method successful across different network architectures / explanation methods at a given budget? How does the perturbation magnitude trade off with the quality of the adversarial example? How does the computational budget? These questions cannot be answered by a handful of pre-selected examples, and are necessary to validate the authors' claims about the method. While the authors include several figures in the appendix, a concise summary should be included in the main body of the paper.

  -  The method is very sample-expensive ( using 50K model queries per example). Part of this can be attributed to the parameterization of the problem that allows for the use of the reinforce/reparameterization trick. This begs the question of why this particular approach was chosen, and I didn't notice a compelling justification for this approach in the paper. In principle, it seems as though using a unimodal gaussian distribution shouldn't offer much benefit over a greedy method that would require fewer model queries per generation.

---

### Decision · Action_Editors · 2023-08-01

**Recommendation:** Accept with minor revision

**Comment:**

All reviewers agree that the paper is interesting and they all voted for acceptance. They raised several issues, please see the list below:
1. Please update the claims about imperceptible perturbations to more accurately reflect the (frequently clearly perceptible) examples shown in the paper.
2. The choice of method would benefit from further justification as to why this particular form of population-based search is needed.
3. Please make appropriate use of \citep and \citet. The sudden appearance of names outside of the citation's parentheses is confusing when the citation is used in passing.
4. The success of the black-box attack on explanation maps will require a sufficient number of queries to generate effective adversaries. It will be constructive if the paper elaborates more details on the query cost (e.g., time of inference/API call) and discusses how adversarial effectiveness is affected by the number of queries.
5. The paper shall provide more details on how the crafted loss can balance two objectives: maintaining the model prediction while manipulating the explanation map.
6. Authors should analyze the perturbation budget (i.e., the numerical range of noise) imposed on the image. A too-large budget will affect the image quality and make it hard to pass user observation. The paper should evaluate how the perturbation affects the image quality and perform an analysis of the relation of the perturbation budget with the attack success rate.

If any of these points are already addressed in the paper, please comment on them.


**Audience:**

The paper is interesting to TMLR's audience. First, XAI is an important line of research. Second, it touches upon adversarial attacks, which is also an intruging research direction.

**Claims And Evidence:**

The paper provides one claim, namely:
- The paper introduces a new black-box algorithm (AttaXAI) that enables manipulation of an image through a barely noticeable perturbation, without the use of any model internals, such that the explanation fits any given target explanation.

This claim is further evaluated empirically on 2 benchmark datasets, using 4 different models, and 5 different XAI methods.

Overall, the claim is very clear and the provided evidence is sufficient.